# Dynamic enhancer partitioning instructs activation of a growth-related gene during exit from naïve pluripotency

Maxim Greenberg[1]*, Aurélie Teissandier[1], Marius Walter[1†], Daan Noordermeer[2], Deborah Bourc'his[1]*

[1]Institut Curie, PSL Research University, INSERM, CNRS, Paris, France; [2]Institute for Integrative Biology of the Cell (I2BC), Université Paris Sud, Université Paris-Saclay, CEA, CNRS, Paris, France

**Abstract** During early mammalian development, the chromatin landscape undergoes profound transitions. The *Zdbf2* gene—involved in growth control—provides a valuable model to study this window: upon exit from naïve pluripotency and prior to tissue differentiation, it undergoes a switch from a distal to a proximal promoter usage, accompanied by a switch from polycomb to DNA methylation occupancy. Using a mouse embryonic stem cell (ESC) system to mimic this period, we show here that four enhancers contribute to the *Zdbf2* promoter switch, concomitantly with dynamic changes in chromatin architecture. In ESCs, the locus is partitioned to facilitate enhancer contacts with the distal *Zdbf2* promoter. Relieving the partition enhances proximal *Zdbf2* promoter activity, as observed during differentiation or with genetic mutants. Importantly, we show that 3D regulation occurs upstream of the polycomb and DNA methylation pathways. Our study reveals the importance of multi-layered regulatory frameworks to ensure proper spatio-temporal activation of developmentally important genes.
DOI: https://doi.org/10.7554/eLife.44057.001

*For correspondence:
maxim.greenberg@curie.fr (MG);
Deborah.Bourchis@curie.fr (DB)

Present address: †The Buck Institute for Research on Aging, Novato, United States

## Introduction

During the early stages of mammalian development, as the embryo implants into uterine wall, the pluripotent cells that will go on to form somatic tissues transition from 'naïve' to 'primed' for lineage specification (*Nichols and Smith, 2009*). One hallmark of the naïve pluripotent state is globally low DNA methylation, whereas primed cells are highly DNA methylated (*Seisenberger et al., 2012*). Incidentally, chromatin architecture and the underlying histone modifications are also dramatically remodeled during this period (*Lee et al., 2014*; *Xu and Xie, 2018*). Collectively, this process is referred to as epigenetic reprogramming, and it accompanies dynamic changes to the transcriptional landscape.

Yet, during the naïve-to-primed transition, certain genes must maintain constant expression despite the massively reshaping epigenome underfoot. A well-documented mechanism during this window is the phenomenon of enhancer switching: as cells differentiate, and therefore express a different suite of transcription factors, enhancers specific for the naïve state will cease their activity, and enhancers attuned to the primed state will become active. In this manner, distinct enhancers can regulate the same promoter, thus ensuring continuous expression. Notably, the general regulator of pluripotency *Oct4* (also known as *Pou5f1*) relies on an naïve and primed cell-specific enhancers (*Tesar et al., 2007*; *Yeom et al., 1996*), as does the developmental regulator *Nodal* (*Papanayotou et al., 2014*). Recently, it was demonstrated that the transcription factor GRHL2 coordinates a large set of these enhancer switches for epithelial genes by activating the primed set of enhancers during differentiation (*Chen et al., 2018*).

The *Zdbf2* locus exhibits an alternative strategy to maintain expression during this cellular transition. In the naïve state, the distal *Long isoform of Zdbf2* promoter (*pLiz*) is active. As cells exit naïve pluripotency, a promoter switch occurs, resulting in activation of the proximal *Zdbf2* promoter (*pZdbf2*), located 73 kilobases (kb) downstream. From the primed state and then throughout life, *pZdbf2* is the functional promoter while *pLiz* is constitutively silenced. It should be noted that despite the genomic distance, there is no change in the message between the *Liz* and *Zdbf2* isoforms, thus the promoter switch does not result in protein diversity. Rather, there is a stratified relationship between the two promoters: *pLiz* activity is absolutely required for imprinted deposition of DNA methylation at a somatic differentially methylated region (sDMR). The sDMR DNA methylation, in turn, antagonizes a block of polycomb-mediated repression, freeing *pZdbf2* (*Figure 1—figure supplement 1A*) (*Duffié et al., 2014*; *Greenberg et al., 2017*). Mice that are deficient for *pLiz* are never able to activate *pZdbf2*, and this leads to a substantial growth defect. Thus, the *Zdbf2* locus provides a valuable model to dissect how promoter switching occurs in concert with changing chromatin dynamics during cellular differentiation.

We show here using a cell-based approach that several enhancers cooperate to regulate the dynamics activity of the different *Zdbf2* promoters. Moreover, CCCTC-BINDING FACTOR (CTCF)-mediated contacts at the locus change dynamically during differentiation, and contribute to the sequential activity of the *Zdbf2* promoters. Saliently, 3D organization of the locus appears to exert its effect epistatically with respect to the DNA methylation and polycomb pathways. This implies that there are two layers of chromatin-based control of *Zdbf2*: one at the level of chromatin marks and the other at the level of chromatin architecture. The highly regulated nature of *Zdbf2* underscores the importance of structural chromosome topology occurring in concert with chromatin marks to control proper spatio-temporal expression of developmentally consequential genes.

## Results

### Two classes of putative enhancers lie in the *Zdbf2* locus

To discover functional genetic elements that regulate *Zdbf2* alternative promoter usage during de novo DNA methylation, we performed an assay for transposase-accessible chromatin followed by sequencing (ATAC-Seq) (*Buenrostro et al., 2013*) on DNA hypomethylated naïve ESCs (cultured in 2i/LIF + vitC) and on primed, highly DNA methylated epiblast-like cells (EpiLCs) at day 7 (D7) of differentiation (*Figure 1A*). A protracted differentiation protocol is necessary to observe the promoter-switch dynamics (*Figure 1—figure supplement 1B*) (*Greenberg et al., 2017*), which distinguishes this protocol from typical short-term EpiLC differentiation methods that generally last two or three days. As such, the transcriptome at later time-points of differentiation is more in line with primed epiblast stem cells (EpiSCs) in culture, as opposed to 'formative' EpiLCs (*Figure 1—figure supplement 1C,D*) (*Bao et al., 2018*; *Kalkan and Smith, 2014*). For clarification, we therefore refer to these cells as D7 EpiLCs. In this cellular system, the imprinted status of the locus is lost, but it biallelically recapitulates all events occurring in vivo on the paternal allele, including the *pLiz* to *pZdbf2* promoter usage switch and sDMR methylation (*Greenberg et al., 2017*). As expected, the ATAC-Seq peak for *pLiz* diminished as it became repressed and DNA methylated from ESCs to D7 EpiLCs. An ATAC-Seq peak was present at *pZdbf2* in ESCs and further enhanced in D7 EpiLCs; this is correlated with our previous data indicating that *pZdbf2* is bivalent and poised in ESCs (*Greenberg et al., 2017*; *Mas et al., 2018*).

In between the two promoters, four significant peaks were present in both ESCs and D7 EpiLCs, three proximal to *pLiz* (E1-3), and one adjacent to a CpG island (CGI) that is an apparent border to the polycomb H3K27me3 block in ESCs (E4) (*Figure 1A*). Given that these regions of accessible chromatin were not lying on obvious active promoters, we reasoned that they were potential enhancer elements and named them E1 to E4, from the closest to the most distal to *pLiz*. Therefore we assayed for enrichment of H3K27 acetylation (H3K27ac), a mark of active enhancer elements (*Creyghton et al., 2010*). E1-3 appeared enriched for the H3K27ac mark in both cell types, while E4 was enriched for the mark only in D7 EpiLCs (*Figure 1B*). Thus, while E1-3 can be classified as active in ESCs and EpiLCs, the chromatin accessibility and H3K27ac dynamics at E4 are reminiscent of so-called 'poised' enhancers (*Figure 1—figure supplement 2A*) (*Buecker et al., 2014*; *Rada-Iglesias et al., 2011*). Moreover, publicly available data indicate that E4 is marked by P300 in ESCs

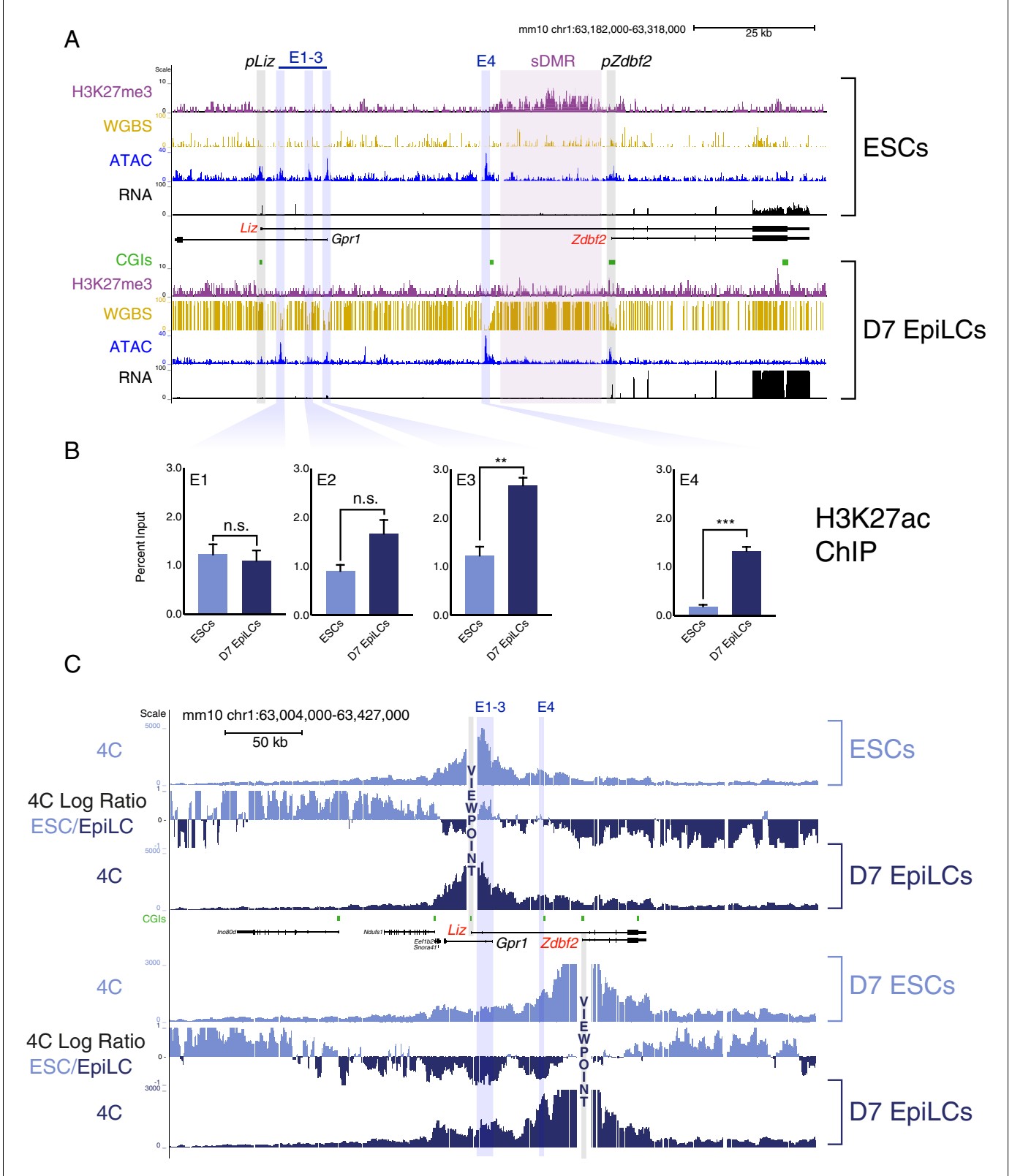

**Figure 1.** The *Zdbf2* locus exhibits dynamic enhancer activity during differentiation. (**A**) Chromatin and expression landscape at the *Zdbf2* locus in ESCs (top) and D7 EpiLCs (bottom). In hypomethylated naïve ESCs, *Zdbf2* initiates from the *pLiz* promoter while a ~ 25 kb block of H3K27me3 extends through *pZdbf2*. Upon EpiLC differentiation, the H3K27me3 signal depletes while DNA methylation is gained; the *pLiz* ATAC-seq peak decreases concomitantly with decreased expression, while *pZdbf2* becomes the main promoter. Four prominent ATAC-seq peaks of accessible chromatin (E1 to

*Figure 1 continued on next page*

*Figure 1 continued*

E4) lie between *pLiz* and *pZdbf2* promoters. WGBS: Whole genome bisulfite sequencing. H3K27me3 ChIP-seq data is from *Greenberg et al. (2017)*. All other genomics data were generated for this study. One representative biological replicate is displayed for each RNA-seq track. (B) H3K27ac ChIP-qPCR shows enrichment for this mark at E1 to E3 in ESCs and D7 EpiLCs, while E4 only becomes enriched in EpiLCs. Data shown as ±s.e.m. from three biological replicates. (C) 4C-seq tracks from the *pLiz* (top) and *pZdbf2* (bottom) viewpoints (VPs) in ESCs and D7 EpiLCs. ESC/EpiLC ratio between 4C-seq signals is indicated. *pLiz* interactions with E1-3 are more frequent in ESCS, but remain high in EpiLCs, likely due to proximity. *pLiz* seems less restricted in EpiLCs, with interactions spreading on the 'right' side of the locus. *pZdbf2* exhibits increased interactions at E1 to E4 in EpiLCs over ESCs. Data from one representative biological replicate (two total). Statistical analyses were performed by two-tailed unpaired t-test: n.s = not significant, **p≤0.01, ***p≤0.001.

DOI: https://doi.org/10.7554/eLife.44057.002

The following source data and figure supplements are available for figure 1:

**Source data 1.** Source data for *Figure 1*.
DOI: https://doi.org/10.7554/eLife.44057.006
**Figure supplement 1.** *Zdbf2* expression dynamics during differentiation.
DOI: https://doi.org/10.7554/eLife.44057.003
**Figure supplement 2.** Chromatin landscape of the *Zdbf2* locus *in cellula* and in vivo.
DOI: https://doi.org/10.7554/eLife.44057.004
**Figure supplement 3.** *Zdbf2* locus interactions restricted within inter-TAD region.
DOI: https://doi.org/10.7554/eLife.44057.005

and shows high levels of vertebrate conservation (*Figure 1—figure supplement 2B*), two more features of poised enhancers (*Rada-Iglesias et al., 2011*). OCT4 is enriched at all of the putative enhancers in both naïve ESCs and D7 EpiLCs (two pluripotent cell types), indicating that both classes of enhancers are likely regulated in a pluripotency-dependent manner (*Buecker et al., 2014*). Importantly, publicly available in vivo data from the naïve pluripotent inner cell mass (ICM) of the blastocyst exhibit a chromatin accessibility and H3K27me3 pattern akin to what we observed in ESCs for the *Zdbf2* locus, suggesting that the in vivo and in cellula regulation are coherent (*Figure 1—figure supplement 2B*) (*Liu et al., 2016*; *Wu et al., 2016*).

## High resolution 4C reveals enhancer-promoter dynamic interactions

Given that E1-4 exhibit the chromatin signature of enhancer elements, it is possible that the *Liz* and *Zdbf2* promoters undergo dynamic interactions with these regulatory elements during the ESC to EpiLC transition. To test this, we performed high-resolution circular chromosome conformation capture followed by sequencing (4C-seq) during differentiation (*Noordermeer et al., 2011*; *van de Werken et al., 2012*). Mammalian genomes are physically subdivided into 'regulatory neighborhoods' known as topologically associated domains (TADs), which average roughly one megabase in size (*Dixon et al., 2012*; *Nora et al., 2012*); available Hi-C data from mouse ESCs indicates that the *Zdbf2* locus exists within an 'inter-TAD' that spans ~650 kb (*Figure 1—figure supplement 3A*) (*Bonev et al., 2017*). Our 4C-seq data further allowed us to subdivide this inter-TAD, with intra-*Zdbf2* locus and intra-*Adam23* locus interactions occurring in relatively mutually exclusive domains (*Figure 1—figure supplement 3B*).

Using the *pLiz* as a 4C-seq viewpoint (VP), we did not observe distal looping that occurred at high frequency in either ESCs or D7 EpiLCs (*Figure 1C*). However, in ESCs, when *Liz* is expressed, the *pLiz* promoter did exhibit increased interactions with the E1-3 cluster relative to EpiLCs, all of which are marked by H3K27ac in this cell type (*Figure 1B*). This is consistent with the possibility that E1-3 contribute to *pLiz* regulation. Given the close proximity between *pLiz* and E1-3, interactions remained unsurprisingly high in EpiLCs, when *Liz* is repressed. No marked looping appeared to occur between *pLiz* and E4 in either ESCs or D7 EpiLCs.

A clear picture emerged from the analysis for the *Zdbf2* promoter (*pZdbf2*), which is active in D7 EpiLCs while E1-4 are all marked by H3K27ac. Our 4C-seq revealed that *pZdbf2* exhibited increased contacts with all four of the putative enhancers in D7 EpiLCs (*Figure 1C*). This indicates a potential cooperative role for E1-4 in activating *pZdbf2*.

## Determination of enhancer function and regulation

From our 4C-seq analyses we reasoned that E1-3 potentially regulate *pLiz* in ESCs, while E4 is inactive (*Figure 2A*). To test this, we generated homozygous deletions of combinations of putative enhancer elements (*Figure 2—figure supplement 1A*). The E3 element also serves as the promoter of the *Gpr1* gene, which is lowly expressed in our system and we previously showed plays no role in *Zdbf2* regulation (*Greenberg et al., 2017*). As such, deleting the element had no impact on expression or DNA methylation at the *Zdbf2* locus (*Figure 2—figure supplement 1B,C*). If E3 is an enhancer element, it may be redundant with E1 and/or E2. Therefore, we generated a ~ 13 kb deletion that encompassed E1-3 (*Figure 2B*, *Figure 2—figure supplement 1A*). In the absence of these elements, the *Liz* transcript was markedly repressed, and the canonical *Zdbf2* isoform failed to properly activate (*Figure 2C*). As *Liz* transcription is required to activate *pZdbf2*, it should be noted that this deletion does not confirm E1-3 elements regulate *pZdbf2* directly. However, the data provide a strong indication that E1-3 are indeed enhancers of *pLiz*.

We previously showed that DNA methylation accumulates at *pLiz* after *Liz* transcription ablates (*Greenberg et al., 2017*). Interestingly, in the absence of E1-3, DNA methylation accumulated faster at *pLiz*, perhaps indicating less protection from de novo DNA methyltransferases due to reduced transcription factor occupancy (*Figure 2D*). *Liz* transcription is required for *cis* DNA methylation establishment at the sDMR region in cellula and in vivo (*Greenberg et al., 2017*). In the absence of E1-3, DNA methylation failed to properly accumulate at the sDMR region, reaching 67% by D7 (*Figure 2D*). This was likely as a consequence of reduced *Liz* expression, in agreement with the 45% sDMR methylation we previously reported upon complete deletion of *pLiz* (*ΔLiz*) (*Greenberg et al., 2017*).

Upon the *pLiz*-to-*pZdbf2* promoter switch, E4 becomes enriched for H3K27ac. We hypothesized that a deletion for E4 would have minimal impact on *pLiz*, but may affect *pZdbf2* activity (*Figure 2A, E*). Indeed, *ΔE4* mutant cells exhibited no alteration of *Liz* expression, but *Zdbf2* transcripts were strongly reduced (*Figure 2F*). As *Liz* was unaffected, there was no impact on DNA methylation at the locus (*Figure 2G*). Moreover, reduced expression of *Zdbf2* in *ΔE4* EpiLCs did not correlate with maintained polycomb occupancy in the sDMR region and *pZdbf2* (*Figure 2H*). In sum, the enhancer E4 is necessary for *pZdbf2* activation, irrespective of the local DNA methylation or polycomb status.

The E4 enhancer element bears the hallmark of a poised enhancer in that it is enriched for P300 in ESCs, but only becomes active in EpiLCs. However, poised enhancers were originally defined as being enriched for H3K27me3 (*Rada-Iglesias et al., 2011*), whereas E4 is depleted for the mark (*Figure 2H*). H3K27me2, which like H3K27me3 is deposited by polycomb repressive complex 2 (PRC2), has also been reported to prevent firing of enhancers in ESCs (*Ferrari et al., 2014*). Yet H3K27me2 ChIP analysis revealed that E4 is relatively depleted for this mark as well (*Figure 2—figure supplement 1D*). E4 does seem to play a role in preventing ectopic polycomb spreading: deleting E4 resulted in a slight increase of H3K27me3 enrichment 1 kb upstream of the WT polycomb domain, however the signal was identical to WT levels by 5 kb upstream (*Figure 2H*).

We previously showed that in ESCs containing loss-of-function mutations in the *Embryonic ectoderm development* (*Eed*) gene (*Schoeftner et al., 2006*)—a core component of PRC2—there was precocious activation of *pZdbf2* (*Greenberg et al., 2017*). Therefore, we wanted to observe if a PRC2 mutant would result in a change in the chromatin status of E4. Indeed, both E4 and *pZdbf2* became enriched for H3K27ac in the absence of polycomb-mediated repression in ESCs (*Figure 3A*). Incidentally, *pLiz* and E1-3, which are already active in ESCs, exhibited no significant change. In *ΔLiz* mutants, *pZdbf2* remains polycomb repressed (*Greenberg et al., 2017*). As such, in the *ΔLiz* mutant, the E4 enhancer did not attain complete levels of H3K27ac during EpiLC differentiation (*Figure 3B*). Thus, while E4 does not display the signatures of direct polycomb regulation, per se, its activity is controlled in a polycomb-dependent manner.

## *Liz* transcription and polycomb play a minor role in 3D organization of the locus

During differentiation, the transcription initiated from *pLiz* and traversing the locus is required for polycomb-to-DNA methylation switch, and *pZdbf2* activation (*Greenberg et al., 2017*). However, our 4C-seq analysis revealed that in the absence of *Liz* transcription, there is only a minor effect on the distal interaction landscape of *pZdbf2* (*Figure 3—figure supplement 1*). Moreover, in *ΔLiz*

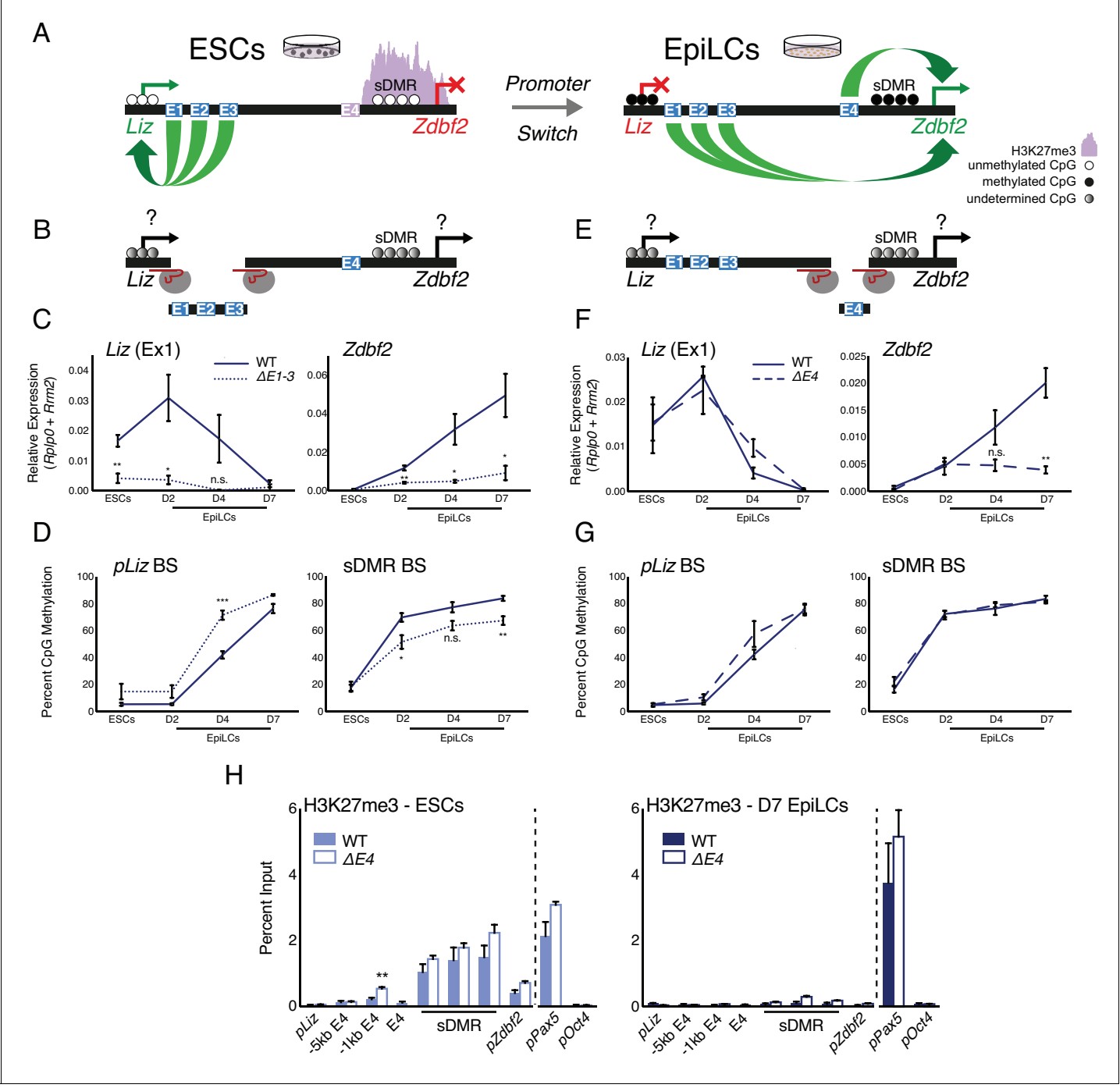

**Figure 2.** Genetic deletions of enhancer elements impact *pLiz/pZdbf2* regulation. (**A**) Model for enhancer regulation based on 4C-seq data. (**B**) Model for deletion of E1-3.. (**C**) RT-qPCR of *Liz* (left) and *Zdbf2* (right) during EpiLC differentiation in WT and the *ΔE1-3* mutant. The *ΔE1-3* mutation significantly reduces both transcripts. Data are shown as ±s.e.m. from five and three biological replicates for WT and mutant, respectively. (**D**) DNA methylation of *pLiz* (left) and the sDMR (right) during EpiLC differentiation as measured by bisulfite conversion followed by pyrosequencing (BS-pyro) in WT and the *ΔE1-3* mutant. When *Liz* fails to activate, DNA methylation is acquired faster at *pLiz*, and fails to properly accumulate at the sDMR. Data shown as ±s.e.m. from five and three biological replicates for WT and mutant, respectively. (**E**) Model for deletion of E4. (**F**) RT-qPCR of *Liz* (left) and *Zdbf2* (right) during EpiLC differentiation in WT and the *ΔE4* mutant. There is no effect on *Liz* expression dynamics, but *Zdbf2* does not properly activate. Data shown as ±s.e.m. from four biological replicates for each genotype. (**G**) DNA methylation of *pLiz* (left) and the sDMR (right) during EpiLC differentiation as measured by BS-pyro in WT and the *ΔE4* mutant. DNA methylation is unperturbed in the *ΔE4* mutant. Data are shown as ±s.e.m. from four biological replicates for each genotype. (**H**) H3K27me3 ChIP-qPCR in ESCs (left) and EpiLCs (right). There is no significant effect on polycomb dynamics in *ΔE4* mutation, except mild ectopic spreading upstream of the sDMR region. *pPax5* and *pOct4* are positive and negative controls,

*Figure 2 continued on next page*

*Figure 2 continued*

respectively. Data shown as ±s.e.m. from three biological replicates for each genotype. Statistical analyses were performed by two-tailed unpaired
t-test: n.s = not significant, *p≤0.05, **p≤0.01, ***p≤0.001.

DOI: https://doi.org/10.7554/eLife.44057.007

The following source data and figure supplement are available for figure 2:

**Source data 1.** Source data for *Figure 2*.

DOI: https://doi.org/10.7554/eLife.44057.009

**Figure supplement 1.** E3 deletion has minimal effect on locus regulation, E4 is devoid of PRC2 signature.

DOI: https://doi.org/10.7554/eLife.44057.008

EpiLCs, *pZdbf2* exhibited increased interactions with E1-4, but not to the same extent as WT EpiLCs. It should be noted that the Δ*Liz* DNA methylation phenotype is only partial in the cell-based system, which may account for the intermediate chromosome conformation phenotype.

The polycomb region that regulates *pZdbf2* spans ∼ 25 kb, from E4 and into the body of *Zdbf2* (*Figure 1A*). Consistent with previous reports, in ESCs this region forms a tightly packed domain (*Kundu et al., 2017*) (*Figure 1C*). We performed 4C-seq in *Eed* mutant ESCs in order to determine if polycomb impacts the chromosome conformation (*Figure 3C*). In fact, in a PRC2 mutant, *pZdbf2* interacted even more frequently within the domain normally defined by a polycomb block in WT cells. This is likely due to the activation of E4, and increased promoter-enhancer interactions. It has recently been shown that active promoters exhibit increased agitation in the nucleus, leading to a potential increase of promoter-enhancer contacts (*Gu et al., 2018*). Given that *pZdbf2* becomes preciously active in *Eed* mutant ESCs, logic would dictate that it would interact more frequently with E1-3, which are also active. However, our 4C-seq in the polycomb mutant showed that this was not the case (*Figure 3C*). To summarize, *Liz* transcription and the polycomb status play a limited role in the regulation of the *pZdbf2* interaction landscape in ESCs, and there must be other mechanisms in place.

## CTCF partitions the *Zdbf2* locus in ESCs

Given that *pZdbf2* does not interact with E1-3 in polycomb mutant ESCs, those enhancers must be restricted from forming long-range loops. The most likely candidate to contribute to locus organization is CTCF (*Ong and Corces, 2014*). We analyzed the 4C-seq patterns of four CTCF binding sites (*Stadler et al., 2011*) present throughout the locus (data available upon request). In ESCs, a CTCF-binding site proximal to the *Gpr1* promoter formed a looping structure with two CTCF sites downstream of the *Gpr1* gene (*Figure 4A*). Incidentally, *pLiz* and E1-3 co-reside within this genomic segment. During differentiation to EpiLCs, we found that the *pLiz*/E1-3 interactions are reduced. In accordance, CTCF binding at this site depleted, whereas CTCF remained bound at the sites downstream of *Gpr1* (*Figure 4B*). Therefore, we referred to this binding platform as the 'CTCF_partition site' (CTCF_PS), which physically separates the active *pLiz*/E1-3 region from the silent *pZdbf2*/E4 region in ESCs (*Figure 4C*). In EpiLCs, disappearance of CTCF at the partition site would then allow for *pZdbf2* to interact with E1-3, while *pLiz* is silenced.

Using the CTCF_PS as a VP in our *Eed* mutant ESCs, we observed that the partition loop still formed in absence of polycomb (*Figure 4—figure supplement 1A*). Furthermore, CTCF still remained enriched at the CTCF_PS in PRC2 mutant ESCs (*Figure 4B*). The continued formation of the partition in the absence of polycomb-mediated regulation would explain why *pZdbf2* failed to exhibit increased interactions with E1-3, even though the promoter has adopted an active state.

Given that CTCF is DNA methylation sensitive at a subset of binding sites (*Wang et al., 2012*), we reasoned that perhaps de novo DNA methylation is required for evicting CTCF from the CTCF_PS during the ESC to EpiLC transition. We tested this by differentiating *Dnmt* tKO ESCs, which are able to differentiate to a state akin to WT D7 EpiLCs despite a total lack of DNA methylation (*Greenberg et al., 2017*; *Hassan-Zadeh et al., 2017*). However, even in the absence of DNA methylation, CTCF depleted at the partition site (*Figure 4D*, *Figure 4—figure supplement 1B*). A recent study reported that transcription can disrupt CTCF binding and chromatin architecture (*Heinz et al., 2018*), yet we observed reduced CTCF enrichment even in the absence of the *Liz* transcript (*Figure 4D*, *Figure 4—figure supplement 1B*). Therefore, the CTCF depletion at the

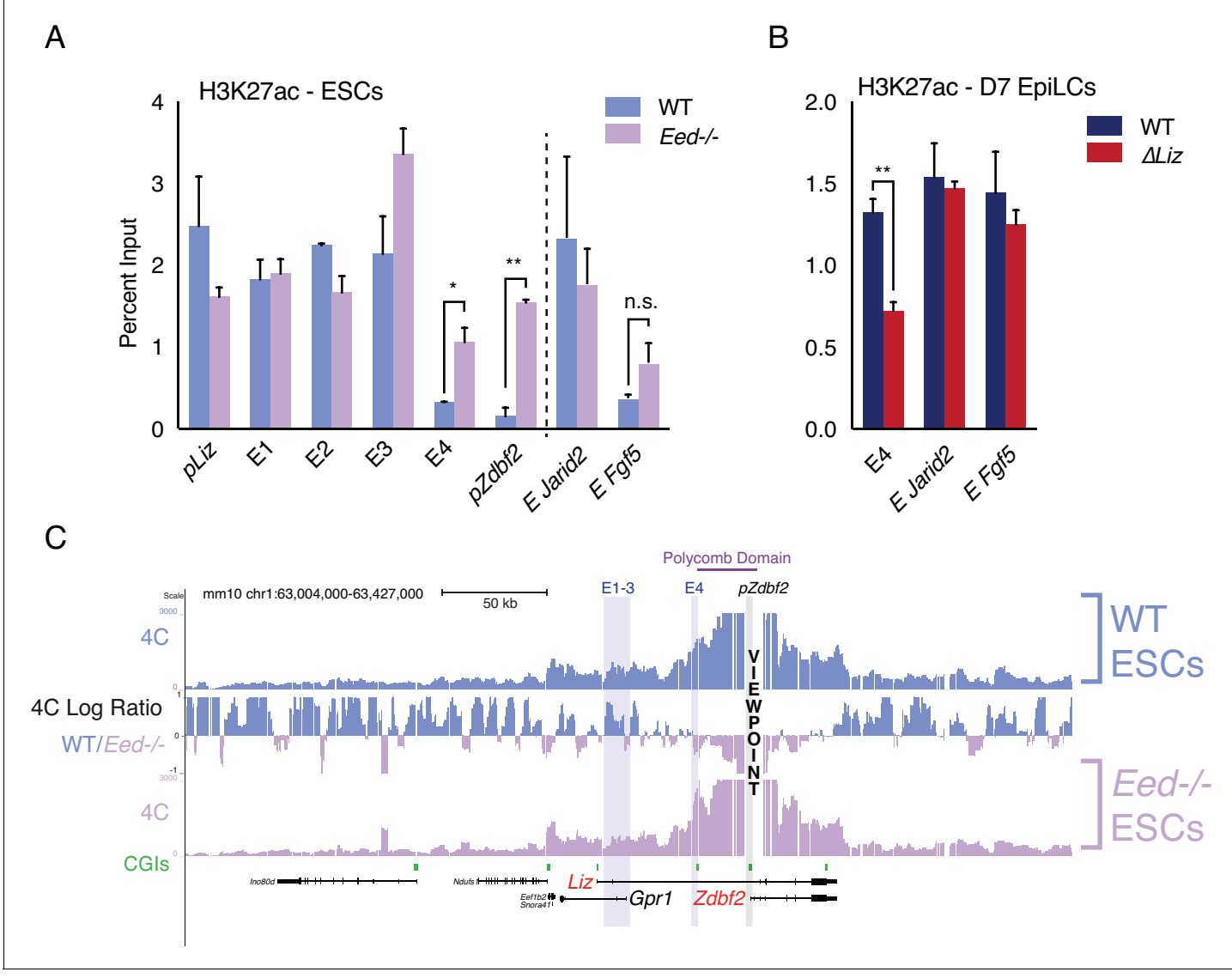

**Figure 3.** Polycomb regulates E4, but plays minor role in chromosome conformation. (A) H3K27ac ChIP-qPCR in WT and *Eed-/-* ESCs. H3K27ac levels are unaffected at *pLiz* and E1-3, but E4 and *pZdbf2* become aberrantly activated in *Eed-/-* ESCs. Data shown as ±s.e.m. from two biological replicates for both genotypes. (B) H3K27ac ChIP-qPCR in WT and *ΔLiz* EpiLCs. In *ΔLiz* mutants, when the sDMR remains enriched for H3K27me3, E4 remains diminished for H3K27ac. Data shown as ±s.e.m. from three biological replicates for both genotypes. (C) 4C-seq tracks from the *pZdbf2* VP in WT and *Eed-/-* ESCs. Ratios between 4C-seq signals is indicated in between the samples. In *Eed* mutants, *pZdbf2* exhibits increased interactions at E4, but not E1-3. Statistical analyses were performed by two-tailed unpaired t-test: n.s = not significant, *p≤0.05, **p≤0.01.

DOI: https://doi.org/10.7554/eLife.44057.010

The following source data and figure supplement are available for figure 3:

**Source data 1.** Source data for *Figure 3*.
DOI: https://doi.org/10.7554/eLife.44057.012

**Figure supplement 1.** *Liz* exerts minimal effect on chromosome conformation at the *Zdbf2* locus 4C-seq tracks from the *pZdbf2* VP in *ΔLiz* ESCs and EpiLCs and WT EpiLCs.
DOI: https://doi.org/10.7554/eLife.44057.011

CTCF_PS in EpiLCs is differentiation-dependent, but independent of DNA methylation- or *Liz* transcription-based regulation.

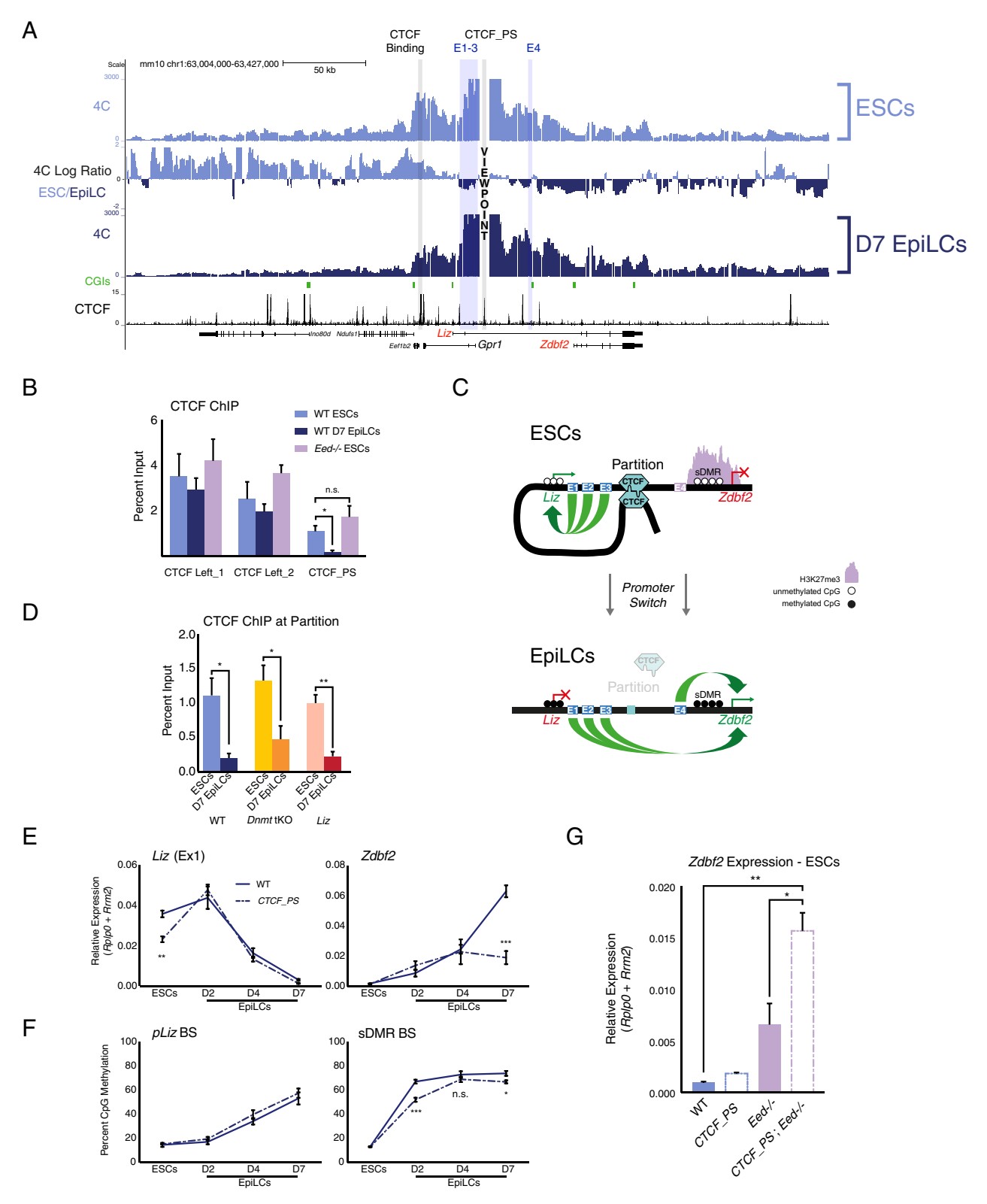

**Figure 4.** The *Zdbf2* locus is partitioned by CTCF, which instructs expression dynamics. (**A**) 4C-seq tracks from the CTCF_PS VP in WT ESCs and EpiLCs. Ratios between 4C-seq signals is indicated between the samples, and gene and CTCF binding tracks (*Stadler et al., 2011*) are below. The CTCF_PS forms a loop with two CTCF sites downstream of the *Gpr1* gene. The looping diminishes in EpiLCs. (**B**) CTCF ChIP-qPCR in WT ESCs and EpiLCs and *Eed-/-* ESCs. CTCF binding remains unchanged in all conditions on the two sites downstream of *Gpr1* (termed CTCF Left_1 and Left_2). At

Figure 4 continued

the CTCF_PS, CTCF binding reduces from WT ESCs to EpiLCs, which is correlated with decreased interactions between CTCF_PS and the two upstream CTCF sites. CTCF binding remains enriched in *Eed-/-* ESCs, consistent with the maintained loop structure. Data shown as ±s.e.m. from three biological replicates. (C) CTCF ChIP-qPCR in WT, *Dnmt* tKO and Δ*Liz* ESCs and EpiLCs. CTCF binding is depleted at the CTCF_PS in EpiLCs in all three contexts. Data shown as ±s.e.m. from three biological replicates. (D) Model for CTCF-mediated partitioning of the locus. In ESCs, *pLiz* and E1-3 are active, and physically separated from the silent E4 and *pZdbf2*. During differentiation, the partition is diminished, allowing E1-3 to bolster *pZdbf2* activation, while *pLiz* has become silent. (E) RT-qPCR of *Liz* (left) and *Zdbf2* (right) during EpiLC differentiation in WT and the Δ*CTCF_PS* mutant. *Liz* is less expressed in mutant ESCs, but reaches WT levels of expression during differentiation. Nevertheless, *Zdbf2* does not properly activate in the mutant. Data are shown as ±s.e.m. from four biological replicates for each genotype. (F) DNA methylation of *pLiz* (left) and the sDMR (right) during EpiLC differentiation as measured by BS-pyro in WT and the Δ mutant. DNA methylation is unperturbed in the mutant at *pLiz*, but is reduced at the sDMR. Data shown as ±s.e.m. from four biological replicates for each genotype. (G) RT-qPCR of *Zdbf2* in ESCs in absence of CTCF partition and/or PRC2. While *Zdbf2* is already upregulated in the *Eed* mutant, this effect is exacerbated in the absence of the partition, likely because *pZdbf2* is less restrained from interacting with E1-3. Data shown as ±s.e.m. from three biological replicates for each genotype. Statistical analyses were performed by two-tailed unpaired t-test: n.s. = not significant, *p≤0.05, **p≤0.01, ***p≤0.001.

DOI: https://doi.org/10.7554/eLife.44057.013

The following source data and figure supplements are available for figure 4:

**Source data 1.** Source data for *Figure 4*.

DOI: https://doi.org/10.7554/eLife.44057.017

**Figure supplement 1.** CTCF dynamics maintained in polycomb, DNA methylation, and *Liz* mutant background.

DOI: https://doi.org/10.7554/eLife.44057.014

**Figure supplement 2.** CTCF_PS deletion impacts locus regulation.

DOI: https://doi.org/10.7554/eLife.44057.015

**Figure supplement 3.** Polycomb and 3D organization both impact *Zdbf2* activation.

DOI: https://doi.org/10.7554/eLife.44057.016

## CTCF partitioning fine-tunes *pLiz* programming of *pZdbf2*

To assess the regulatory impact of CTCF partitioning, we generated a deletion of CTCF_PS (*Figure 4—figure supplement 2A*). 4C-seq in Δ*CTCF_PS* cells revealed that *pLiz* interacts less frequently with E1-3 in ESCs (*Figure 4—figure supplement 2B*), perhaps as the promoter is less constrained without the CTCF partition. As such, *Liz* failed to properly express (*Figure 4E*). In contrast, the deletion did not perturb expression of *Gpr1* nor the genes flanking the *Zdbf2* locus (*Figure 4—figure supplement 2C*), suggesting that insulation from neighboring regulatory domains was intact. Consistently, using a CTCF degron line where CTCF is globally depleted (*Nora et al., 2017*), *Liz* was also downregulated, although it should be noted that both moderate and complete reduction of CTCF resulted in a comparable reduction of *Liz* levels (*Figure 4—figure supplement 2D,E*). During differentiation of the Δ*CTCF_PS* line, the *Liz* transcript was still able to attain WT levels, nevertheless *Zdbf2* failed to properly activate (*Figure 4E*). Moreover, while DNA methylation occurred normally at *pLiz*, the sDMR remained relatively hypomethylated compared to WT (*Figure 4F*), likely due to the delayed kinetics of *Liz* upregulation. Furthermore, the relative reduction of DNA methylation at the sDMR in Δ*CTCF_PS* mutant EpiLCs was correlated with a slight retention of H3K27me3 in comparison with WT, which may contribute to the failure of *pZdbf2* to properly activate (*Figure 4D*, *Figure 4—figure supplement 3A*).

While *pLiz* activation was less efficient in Δ*CTCF_PS* mutant ESCs, *pZdbf2* exhibited increased contacts with E2 and E3 (*Figure 4—figure supplement 2C*). However, *Zdbf2* remained repressed, as polycomb enrichment remained unperturbed (*Figure 4—figure supplement 3A*). It should be noted that upon global depletion of CTCF, *Zdbf2* is de-repressed, indicating that independently of the partition site, CTCF does play a role in *Zdbf2* regulation in ESCs (*Figure 4—figure supplement 2E*). Regardless, given that in PRC2 mutant ESCs, *Zdbf2* is already partially de-repressed, we reasoned that by generating an *Eed* mutation in combination with deleting the partition site, we could observe further increase in *Zdbf2* expression, as *pZdbf2* would be unhindered from interacting with all four enhancers (*Figure 4—figure supplement 3B*). In parallel, we generated a new *Eed* mutation, so all cell lines would be in the identical genetic background (*Figure 4—figure supplement 3B*). We confirmed that the *Eed* mutant lines failed to exhibit EED protein nor H3K27me3 by western blotting (*Figure 4—figure supplement 3C*). Indeed, while *Zdbf2* was upregulated in absence of *Eed* alone, the expression was significantly increased in the Δ*CTCF-PS; Eed-/-* double mutant (*Figure 4G*).

Importantly, these results were confirmed when we subjected WT and ∆*CTCF_PS* lines to a potent PRC2 inhibitor to induce acute depletion of H3K27me3 (*Figure 4—figure supplement 3D,E*). Therefore, we concluded that in addition to contributing to proper *pLiz* activation, the CTCF partition acts as a second level of protection, along with polycomb, to restrain precocious *pZdbf2* firing.

## Discussion

In this study, we revealed the dynamic chromosome conformation of the *Zdbf2* locus that occurs concomitantly with epigenetic programming during differentiation. For proper *Zdbf2* activation, it is imperative to properly trigger *Liz* expression at the time de novo DNA methylation. Here we show that a CTCF-structured loop organization forms at the locus in naïve ESCs. This partition allows for proper regulation of *pLiz*, which in turn can facilitate a local polycomb-to-DNA methylation switch through *Liz* transcription. During differentiation, the partitioning is relaxed, as *pLiz* becomes shut down, thus allowing distal enhancers to bolster *pZdbf2* activation. It is conceivable that E1-3 act in a hierarchical manner with respect to E4 and *pZdbf2*; that is, they are important for activation of the proximal promoter rather than sustaining *Zdbf2* expression, although further testing must be performed to formally confirm his possibility. Deleting the CTCF_PS resulted in reduced *Liz* activation kinetics, but a fairly substantial effect on *Zdbf2* expression. This is consistent with our previously published *Liz* transcriptional interruption, where *Liz* expression and DNA methylation are only moderately affected, but nonetheless *Zdbf2* cannot attain WT levels of activation (*Greenberg et al., 2017*). Such results underscore the sensitivity of *pZdbf2* activity to proper epigenetic programming.

In ∆*CTCF_PS* ESCs, *Zdbf2* transcription did not ectopically occur, even though there were no longer apparent restrictions for *pZdbf2* to interact with the active enhancers E1-3. The explanation for this is the polycomb-mediated silencing that persists over *pZdbf2* in the absence of CTCF_PS. Thus, there are at least two layers of regulation of *Zdbf2* activation that act independently from classical transcription factor control at gene promoters: 1) instructive chromosome conformation allowing for proper *pLiz* and *pZdfb2* activity, and 2) a *Liz*-dependent epigenetic switch to evict polycomb at *pZdbf2*. This tiered model, with chromosome conformation acting upstream, emphasizes the exquisite choreography that can occur to program developmentally important genes during the exit from the naïve pluripotent state. Interestingly, global depletion of CTCF leads to upregulation of *Zdbf2* in ESCs (*Figure 4—figure supplement 2E*). Whether this is due to direct CTCF-mediated regulation at other binding sites in the locus, or is simply an indirect effect, remains to be tested.

One outstanding question is what is the cue that releases CTCF from the partition site during differentiation? The obvious candidate was DNA methylation, and while CTCF binding at the site may indeed be DNA methylation-sensitive, we found that CTCF was released during differentiation, whether the methyl mark was present or not. Another likely explanation is that CTCF is bound in combination with pluripotency-associated transcription factors. While CTCF is generally reported to be largely invariant across mammalian cell types (*Lee et al., 2012*), there are cell type-specific CTCF binding patterns, and roughly 60% occur in a DNA methylation-independent manner (*Wang et al., 2012*).

Our study presents a rare description of two alternative promoters utilizing a shared set of enhancers, but in a CTCF-guided, developmentally timed manner. In ESCs, CTCF is required to facilitate one promoter's interactions (*pLiz*) while restricting the other's (*pZdbf2*). Only upon removal of CTCF binding, the opportunity for *pZdbf2* is created to contact its enhancers. On the face of it, such a mechanism resembles the imprinted *Igf2/H19* locus, where CTCF binding near the *H19* gene on the maternal allele restrict interactions between a shared set of enhancers and the more distal *Igf2* gene (*Murrell et al., 2004*). The *Igf2/H19* locus behaves differently from the *Zdbf2* locus, though, as differential CTCF binding is DNA methylation-dependent and set in the gametes—not dynamically regulated during cellular differentiation. Moreover, *Igf2* and *H19* are two different genes, not isoforms of the same gene.

CTCF has been previously shown to mediate enhancer switching. For example, at the *Hoxd* locus in mice, CTCF facilitates the interactions between the same set of gene promoters but different sets of enhancers depending on the context (*Andrey et al., 2013*). As mentioned, dynamic enhancer switching during the naïve-to-primed differentiation is a common mechanism in mammals to maintain gene expression; however, this phenomenon represents the inverse scenario from *Zdbf2* regulation, as during enhancer switching, promoters remain unchanged.

Promoter switching is a widespread and developmentally important phenomenon (*Davuluri et al., 2008*). Aberrant promoter usage is associated with pathologies, such as cancer (*Sarda et al., 2017*). CTCF is a likely candidate to organize genic three-dimensional structure and protect from aberrant promoter firing, as it does at *Zdbf2*. The *Zdbf2* locus presents a compelling case, because if *pLiz*-to-*pZdbf2* promoter switch does not occur at the proper developmental time, synchronized with the de novo DNA methylation program, *pZdbf2* cannot be activated (*Greenberg et al., 2017*). Notably, the organization of this locus is conserved between mouse and humans, implying the likelihood of a shared regulatory mechanism (*Duffié et al., 2014*). Future studies should continue to shed light on the role that CTCF dynamics play in programming developmentally important promoter activity in the crucial window that precedes somatic tissue formation.

# Materials and methods

## Key resources table

| Reagent type (species) or resource | Designation | Source or reference | Identifier | Additional information |
|---|---|---|---|---|
| Cell Line (*M. musculus*) | E14TG2a (WT) | ATCC | CRL-1821 | |
| Cell Line (*M. musculus*) | E14TG2a_ΔLiz | Bourc'his Lab | | *Greenberg et al., 2017* |
| Cell Line (*M. musculus*) | E14TG2a_ΔE1 | Bourc'his Lab | | *Greenberg et al., 2017* |
| Cell Line (*M. musculus*) | E14TG2a_ΔE1-3 | This study | | CRISPR/Cas9 generated mutant, sgRNA oligos are listed in *Supplementary file 1* |
| Cell Line (*M. musculus*) | E14TG2a_ΔE4 | This study | | CRISPR/Cas9 generated mutant, sgRNA oligos are listed in *Supplementary file 1* |
| Cell Line (*M. musculus*) | E14TG2a_CTCF -AID-eGFP, Tir1 | Gift from E Nora | | *Nora et al., 2017* |
| Cell Line (*M. musculus*) | J1 (WT) | ATCC | SCRC-1010 | |
| Cell Line (*M. musculus*) | J1_Dnmt tKO | Gift from M Okano | | *Tsumura et al., 2006* |
| Cell Line (*M. musculus*) | J1_Eed-/- | This study | | CRISPR/Cas9 generated mutant, sgRNA oligos are listed in *Supplementary file 1* |
| Cell Line (*M. musculus*) | J1_ΔCTCF_PS | This study | | CRISPR/Cas9 generated mutant, sgRNA oligos are listed in *Supplementary file 1* |
| Cell Line (*M. musculus*) | J1_Eed-/-; ΔCTCF_PS | This study | | CRISPR/Cas9 generated mutant, sgRNA oligos are listed in *Supplementary file 1* |
| Cell Line (*M. musculus*) | J1 Clone 36 (WT) | Gift from A Wutz | | *Wutz and Jaenisch, 2000* |
| Cell Line (*M. musculus*) | J1 Clone 36_Eed-/- | Gift from A Wutz | | *Schoeftner et al., 2006* |

*Continued on next page*

*Continued*

| Reagent type (species) or resource | Designation | Source or reference | Identifier | Additional information |
|---|---|---|---|---|
| Antibody | Anti-H3K27me2, mouse monoclonal | Active Motif | 61435 | (1:5000) |
| Antibody | Anti-H3K27me3, rabbit monoclonal | Cell Signaling Technology | C36B11 | (1:5000) |
| Antibody | Anti-H3K27ac, rabbit polyclonal | Active Motif | 39133 | (1:5000) |
| Antibody | Anti-CTCF, rabbit polyclonal | Millipore | 07–729 | (1:5000) |
| Antibody | Anti-GFP, mix of two mouse monoclonal | Roche | 11814460001 | (1:2000) |
| Antibody | Anti-PCNA, mouse monoclonal | Dako | M0879 | (1:5000) |
| Antibody | Anti-EED, rabbit polyclonal | Other | | Gift from R Margueron (1:5000) |
| Antibody | Anti-Lamin B1, rabbit polyclonal | Abcam | Ab16048 | (1:5000) |
| Antibody | Anti-H3, rabbit polyclonal | Abcam | Ab1791 | (1:10000) |
| Chemical Compound, Drug | L-Absorbic Acid | Sigma | A4544 | |
| Chemical Compound, Drug | Gsk3 inhibitor | Other | CT-99021 | Gift from E Heard |
| Chemical Compound, Drug | MEK inhibitor | Other | PD0325901 | Gift from E Heard |
| Chemical Compound, Drug | FGF2 | R and D Systems | 233-FB-025/CF | |
| Chemical Compound, Drug | Activin A | R and D Systems | 338-AC-050/CF | |
| Chemical Compound, Drug | EZH2 Inhibitor | Tocris Bioscience | UNC 1999 | |
| Chemical Compound, Drug | EZH2 Inhibitor Negative Control | Tocris Bioscience | UNC 2400 | |
| Chemical Compound, Drug | Indole-3-acetic acid sodium salt (auxin analog) | Sigma | I5148-2G | |
| Recombinant DNA Reagent | pX459 | Addgene | 62988 | |
| Commercial Assay or Kit | EpiTect Bisulfite Kit | Qiagen | 59104 | |
| Software, Algorithm | BWA v0.7.5a | *Li and Durbin, 2009* | | |
| Software, Algorithm | Picard v1.130 | Broad Institute | | |
| Software, Algorithm | HOMER v4.7 | *Heinz et al., 2010* | | |
| Software, Algorithm | FASTX-Toolkit v0.0.13 | Greg Hannon Lab | | |
| Software, Algorithm | Cutadapt | *Martin, 2011* | | |
| Software, Algorithm | Bismark v0.12.5 | *Krueger and Andrews, 2011* | | |

*Continued on next page*

*Continued*

| Reagent type (species) or resource | Designation | Source or reference | Identifier | Additional information |
|---|---|---|---|---|
| Software, Algorithm | Bowtie2 v2.1.0 | *Langmead and Salzberg, 2012* | | |
| Software, Algorithm | STAR v2.5.0a | *Dobin et al., 2013* | | |
| Software, Algorithm | Trim Galore v0.4.0 | Babraham Institute | | |
| Software, Algorithm | FourCSeq v1.12.0 | *Klein et al., 2015* | | |
| Software, Algorithm | HTSeq v0.9.1 | *Anders et al., 2015* | | |

## ESC lines

All cell lines are listed in the Key Resource Table. For all experiments, the parental WT line was used as a control for mutant lines generated in that background (E14 or J1). 4C-Seq was performed using clone 36 *Eed-/-* cells (J1 background). Therefore, when generating an in-house *Eed-/-* line, we used the same genetic background for consistency.

## Cell culture and differentiation

Feeder-free ESCs were grown on gelatin-coated flasks. Serum culture conditions were as follows: Glasgow medium (Sigma) supplemented with 2 mM L-Glutamine (Gibco), 0.1 mM MEM non-essential amino acids (Gibco), 1 mM sodium pyruvate (Gibco), 15% FBS, 0.1 mM β-mercaptoethanol and 1000 U/ml leukemia inhibitory factor (LIF, Chemicon). Cells were passaged with trypsin replacement enzyme (Gibco) every two days. 2i culture conditions were as follows: N2B27 medium (50% neurobasal medium (Gibco), 50% DMEM/F12 (Gibco), 2 mM L-glutamine (Gibco), 0.1 mM β- mercaptoethanol, NDiff Neuro2 supplement (Millipore), B27 serum-free supplement (Gibco)) supplemented with 1000 U/ml LIF and 2i (3 µM Gsk3 inhibitor CT-99021, 1 µM MEK inhibitor PD0325901). Cells were passaged every 2–4 days with Accutase (Gibco). Vitamin C (Sigma) was added at a final concentration of 100 µg/ml. For EZH2 inhibition experiments, the EZH2 incubator UNC 1999 (or its negative control UNC 2400, Tocris Bioscience) was added to media at a 1 µM final concentration for four days. To induce degradation of CTCF in the E14TG2a_CTCF-AID-eGFP, Tir1 cell line (with Tir1 targeted to the *Tigre* locus), the auxin analog indole-3-acetic acid (IAA) was added to the media at a final concentration of 500 µM from a 1000x stock, and incubated with the cells for two days.

To induce EpiLC differentiation, cells were gently washed with PBS, dissociated, and replated at a density of $2 \times 10^5$ cells/cm$^2$ on Fibronectin (10 µg/ml)-coated plates in N2B27 medium supplemented with 12 ng/ml Fgf2 (R and D) and 20 ng/ml Activin A (R and D). EpiLCs were passaged with Accutase at D4 of differentiation.

Cells were regularly tested for presence of mycoplasma by sending used media to GATC/Eurofins for analysis.

## Generation of edited ESCs

All deletions in this study were generated with two CRISPR single guide RNAs (sgRNAs) specific to the target sequences followed by Cas9 nuclease activity and screening for non-homologous end joining. sgRNAs were designed using the online CRISPOR online program (crispor.tefor.net) and cloned into the pX459 plasmid harboring the *Cas9* gene. All sgRNA sequences are listed in *Supplementary file 1*. Around five million WT serum-grown ESCs were transfected with 1–3 µg of plasmid(s) using Amaxa 4d Nucleofector (Lonza) and plated at a low density. Ninety-six individual clones were picked and screened by PCR. Mutated alleles were confirmed by Sanger sequencing of cloned PCR amplicons. In the case of the *Eed* mutation, loss-of-function was further confirmed by immunoblotting.

## DNA methylation analyses

Genomic DNA from cells was isolated using the GenElute Mammalian Genomic DNA Miniprep Kit (Sigma), with RNase treatment. Bisulfite conversion was performed on 500–1000 ng of DNA using the EpiTect Bisulfite Kit (Qiagen). Bisulfite-treated DNA was PCR amplified and analyzed by pyrosequencing. Pyrosequencing was performed on the PyroMark Q24 (Qiagen) according to the manufacturer's instructions, and results were analyzed with the associated software. All bisulfite primers are listed in *Supplementary file 1*. Statistical analyses were performed by a two-tailed unpaired *t-test* using GraphPad Prism6 software.

WGBS data from ESCs were previously generated (*Walter et al., 2016*) and EpiLCs were prepared from 50 ng of bisulfite-converted genomic DNA using the EpiGnome/Truseq DNA Methylation Kit (Illumina) following the manufacturer instructions. Sequencing was performed in 100pb paired-end reads at a 30X coverage using the Illumina HiSeq2000 platform.

## ATAC-seq

ATAC-Seq was performed as described in *Buenrostro et al. (2015)* with minor modifications. Briefly, 50,000 cells were washed, but not lysed. Cells were transposed using the Nextera DNA library prep kit (Illumina) for 30 min at 37˚. DNA was immediately purified using Qiagen MinElute Kit (Qiagen). qPCR was used to determine the optimal cycle number for library amplification. The libraries were sequenced on the Illumina HiSeq2500 platform to obtain $2 \times 100$ paired-end reads.

## Chromatin immunoprecipitation (ChIP)

ChIP was performed exactly as described in *Walter et al. (2016)*. Briefly, cells were cross-linked directly in 15 cm culture plates with 1% formaldehyde. After quenching with 0.125 M glycine, cells were washed in PBS and pelleted. After a three-step lysis, chromatin was sonicated with a Bioruptor (Diagenode) to reach a fragment size averaging 200 bp. Chromatin corresponding to 10 μg of DNA was incubated rotating overnight at 4˚C with 3–5 μg of antibody. A fraction of chromatin extracts (5%) were taken aside for inputs. Antibody-bound chromatin was recovered using Protein G Agarose Columns (Active Motif). The antibody-chromatin mix was incubated on column for 4 hr, washed eight times with modified RIPA buffer and chromatin eluted with pre-warmed TE-SDS (50 mM Tris pH 8.0, 10 mM EDTA, 1% SDS). ChIP-enriched and input samples were reverse cross-linked (65˚C overnight) and treated with RNase A and proteinase K. DNA was extracted with phenol/chloroform/isoamyl alcohol, precipitated with glycogen in sodium acetate and ethanol and resuspended in tris-buffered water. Enrichment compared to input was analyzed by qPCR (Viia7 thermal cycling system, Applied Biosystems). Primers are listed in *Supplementary file 1*.

## 4C-seq

The design of VPs and preparation of 4C-seq libraries was performed as described by *Matelot and Noordermeer (2016)*, with only minor modifications. *DpnII* or its isoschizomer *MboII* (New England Biolabs) were chosen as the primary restriction enzyme, and *NlaIII* (New England Biolabs) as the secondary restriction enzyme. ESC and EpiLC material were harvested from 150 cm$^2$ culture flasks (TPP Techno Plastic Products AG), which provided ample material for up to four technical replicates presuming cells were healthy and near confluency. To avoid technical artifacts, crosslinking and library preparation were performed in parallel for each experiment. For each VP, approximately 1 μg of library material was amplified using 16 individual PCR reactions with inverse primers containing indexed Illumina TruSeq adapters (primer sequences are listed in *Supplementary file 1*). PCR products were originally purified using the MinElute PCR purification kit (Qiagen) to remove unincorporated primer, but we found that purification was more efficiently performed using Agencourt AMPure XP beads (Beckman Coulter). Sequencing was performed on the Illumina NextSeq 500 system, using 75 bp single-end reads with up to 14 VPs multiplexed per run.

## RNA expression

Total RNA was extracted using Trizol (Life Technologies), then DNase-treated and column purified (Qiagen RNeasy Kit). To generate cDNA, RNA was reverse transcribed with SuperscriptIII (Life Technologies) primed with random hexamers. RT-qPCR was performed using the SYBR Green Master Mix on the Viia7 thermal cycling system (Applied Biosystems). Relative expression levels were

normalized to the geometric mean of the Ct for housekeeping genes *Rrm2* and *Rplp0* with the ΔΔCt method. Primers are listed in *Supplementary file 1*. Statistical analyses were performed by a two-tailed unpaired t-test using GraphPad Prism6 software.

RNA-seq libraries were prepared from 500 ng of DNase-treated RNA with the TruSeq Stranded mRNA kit (Illumina). Sequencing was performed in 100pb paired-end reads using the Illumina HiSeq2500 platform.

## Immunoblotting

Western blots were visualized using the ChemiDoc MP (Biorad). The antibodies are listed in the Key Resource Table.

## Quantification and statistical analysis

### ATAC-seq analysis

2 × 100 bp paired-end reads were aligned onto the Mouse reference genome (mm10) using Bwa mem v0.7.5a (*Li and Durbin, 2009*) with default parameters. Duplicate reads were removed using Picard v1.130 (http://broadinstitute.github.io/picard/). Tracks were created using HOMER software v4.7 (*Heinz et al., 2010*).

### WGBS analysis

Whole-genome bisulfite sequencing data were analyzed as described in *Walter et al. (2016)*. Briefly, the first eight base pairs of the reads were trimmed using FASTX-Toolkit v0.0.13:

(http://hannonlab.cshl.edu/fastx_toolkit/index.html). Adapter sequences were removed with Cutadapt v1.3 (*Martin, 2011*) and reads shorter than 16 bp were discarded. Cleaned sequences were aligned onto the mouse reference genome (mm10) using Bismark v0.12.5 (*Krueger and Andrews, 2011*) with Bowtie2-2.1.0 (*Langmead and Salzberg, 2012*) and default parameters. Only reads mapping uniquely on the genome were conserved. Methylation calls were extracted after duplicate removal. Only CG dinucleotides covered by a minimum of 5 reads were conserved.

### RNA-seq analysis

2 × 100 bp paired-end reads were mapped onto the mouse reference genome (mm10) using STAR v2.5.0a (*Dobin et al., 2013*) reporting unique alignments and allowing at most six mismatches per fragment. Tracks were created using HOMER software v4.7 (*Heinz et al., 2010*). Gene-scaled quantification was performed with HTSeq v0.9.1 (*Anders et al., 2015*).

### 4C-seq analysis

Adapters were first trimmed using Trim Galore: v0.4.0, https://www.bioinformatics.babraham.ac.uk/projects/trim_galore/.

Samples were demultiplexed using the script provided with the FourCSeq R package (v1.12.0) (*Klein et al., 2015*). Inverse primer sequences were removed on the 3' end of the reads using Cutadapt v1.12 (*Martin, 2011*). Reads shorter than 15 bp were discarded. Cleaned sequences were aligned onto the mouse reference genome (mm10) using Bowtie2 v2.1.0 (*Langmead and Salzberg, 2012*) allowing one mismatch in the seed (22 bp) and an end-to-end alignment. Subsequent steps were performed using the FourCSeq R package (v1.12.0). The mouse reference genome was in-silico digested using the two restriction enzymes. Restriction fragments that did not contain a cutting site of the second restriction enzyme or are smaller than 20 bp were filtered out. Fragments 2.5 kb up- and downstream from the viewpoint were excluded during the procedure. Intrachromosomal contacts were kept. Valid fragments were quantified. The fragment counts were then normalized per one million reads. Data were smoothed using a running mean function with five informative fragments.

## Data resources

Raw and processed sequencing data reported in this paper have been submitted to GEO, accession number GSE121405.

## Acknowledgments

We would like to thank members of the Bourc'his laboratory and N Servant for insightful experimental and conceptual input, E Nora and B Bruneau for sharing the CTCF Degron ESC line, R Oakey and D Holoch for reviewing the manuscript and making helpful comments and V Piras for sharing reanalyzed Hi-C matrices. We acknowledge the high-throughput sequencing facility of I2BC for its sequencing and bioinformatics expertise and for its contribution to this study for the 4C-seq (https://www.i2bc.paris-saclay.fr/spip.php?article399). Other NGS approaches were carried out by the ICGex NGS platform of the Institut Curie, supported by grants from the ANR-10-EQPX-03 (Equipex) and ANR-10-INBS-09–08 (France Génomique Consortium) from the Agence Nationale de la Recherche ('Investissements d'Avenir' program), the Canceropole Ile-de-France and the SiRIC-Curie program-Grant "INCa-DGOS- 4654. The laboratory of DB is part of the Laboratoire d'Excellence (LABEX) entitled DEEP (11-LBX0044). This research was supported by the ERC (grant ERC-Cog EpiRepro). MVCG was supported by ARC and EMBO (LTF 457–2013) postdoctoral fellowships.

## Additional information

### Competing interests

Deborah Bourc'his: Reviewing editor, *eLife*. The other authors declare that no competing interests exist.

### Funding

| Funder | Grant reference number | Author |
| --- | --- | --- |
| H2020 European Research Council | Cog EpiRepro | Deborah Bourc'his |
| European Molecular Biology Organization | LTF 457-2013 | Maxim Greenberg |
| Fondation ARC pour la Recherche sur le Cancer | Post-doc Fellowship | Maxim Greenberg |

The funders had no role in study design, data collection and interpretation, or the decision to submit the work for publication.

### Author contributions

Maxim Greenberg, Conceptualization, Methodology, Investigation, Writing—original draft, Writing—review and editing; Aurélie Teissandier, Formal analysis; Marius Walter, Investigation; Daan Noordermeer, Methodology, Investigation, Writing—review and editing; Deborah Bourc'his, Conceptualization, Supervision, Funding acquisition, Methodology, Writing—review and editing

### Author ORCIDs

Maxim Greenberg ⓘ https://orcid.org/0000-0001-9935-8763
Daan Noordermeer ⓘ https://orcid.org/0000-0002-9296-7002
Deborah Bourc'his ⓘ https://orcid.org/0000-0001-9499-7291

### Decision letter and Author response

Decision letter https://doi.org/10.7554/eLife.44057.035
Author response https://doi.org/10.7554/eLife.44057.036

## Additional files

### Supplementary files

• Supplementary file 1. List of oligonucleotides used in this study.
DOI: https://doi.org/10.7554/eLife.44057.018

• Transparent reporting form

DOI: https://doi.org/10.7554/eLife.44057.019

## Data availability

Sequencing data have been deposited in GEO under accession code GSE121405.

The following dataset was generated:

| Author(s) | Year | Dataset title | Dataset URL | Database and Identifier |
|---|---|---|---|---|
| Greenberg MVC, Teissandier A, Walter M, Noordermeer D, Bourc'his D | 2019 | Sequencing data from Dynamic enhancer partitioning instructs activation of a growth-related gene during exit from naïve pluripotency | https://www.ncbi.nlm.nih.gov/geo/query/acc.cgi?acc=GSE121405 | NCBI Gene Expression Omnibus, GSE121405 |

The following previously published datasets were used:

| Author(s) | Year | Dataset title | Dataset URL | Database and Identifier |
|---|---|---|---|---|
| Greenberg MV, Glaser J, Borsos M, Marjou FE | 2017 | Liz transcription is required for evicting H3K27me3 marks at the Gpr1/Zdbf2 locus through the deposition of DNA methylation marks. | https://www.ncbi.nlm.nih.gov/geo/query/acc.cgi?acc=GSE79549 | NCBI Gene Expression Omnibus, GSE79549 |
| Walter M, Teissandier A, Pérez-Palacios R, Bourc'his D | 2016 | An epigenetic switch ensures transposon repression upon acute loss of DNA methylation in ES cells | https://www.ncbi.nlm.nih.gov/geo/query/acc.cgi?acc=GSE71593 | NCBI Gene Expression Omnibus, GSE71593 |
| Wu J, Huang B, Chen H, Yin Q | 2016 | The landscape of accessible chromatin in mammalian pre-implantation embryos | https://www.ncbi.nlm.nih.gov/geo/query/acc.cgi?acc=GSE66390 | NCBI Gene Expression Omnibus, GSE66390 |
| Liu X, Wang C, Liu W | 2016 | Distinct features in establishing H3K4me3 and H3K27me3 in pre-implantation embryos | https://www.ncbi.nlm.nih.gov/geo/query/acc.cgi?acc=GSE73952 | NCBI Gene Expression Omnibus, GSE73952 |
| Buecker C, Srinivasan R, Wu Z | 2014 | Reorganization of enhancer patterns in transition from naïve to primed pluripotency | https://www.ncbi.nlm.nih.gov/geo/query/acc.cgi?acc=GSE56138 | NCBI Gene Expression Omnibus, GSE56138 |
| Bao S, Tang WW, Wu B | 2017 | Enhancing the Potency of Mouse Embryonic Stem Cells | https://www.ncbi.nlm.nih.gov/geo/query/acc.cgi?acc=GSE99494 | NCBI Gene Expression Omnibus, GSE99494 |

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
