## [Decision Letter]

Thank you for submitting your article "Dynamic enhancer partitioning instructs activation of a growth-related gene during exit from naïve pluripotency" for consideration by *eLife*. Your article has been reviewed by three peer reviewers, including Michael Buszczak as the Reviewing Editor and Reviewer #1, and the evaluation has been overseen by Kevin Struhl as the Senior Editor. The following individual involved in review of your submission has agreed to reveal their identity:; Alvaro Rada-Iglesias (Reviewer #2).

The reviewers have discussed the reviews with one another and the Reviewing Editor has drafted this decision to help you prepare a revised submission.

Summary:

Recent studies have begun to reveal how specific transcription factors (TFs) serve to change the global chromatin landscape as naïve pluripotent embryonic stem cells transition to a primed pluripotent epiblast-like state. This study builds upon these findings and explores the cause and effect relationship between changes in 3D organization and changes in enhancer activity during this developmental transition. As a model, the authors focus on the regulation of the *Zdbf2* locus, which undergoes a switch from distal (*pLiz*) to proximal (*pZdbf2*) promoter usage, accompanied by a switch in polycomb to DNA methylation occupancy. The authors define a number of different enhancer elements (E1-4). E1-3 are active in ESCs, while E4 becomes active in EpiLCs. The authors use high resolution 4C to show enhancer-promoter dynamic interactions during the process of differentiation. They then delete specific enhancers to demonstrate their functional significance in regards to the regulation of chromatin marks throughout the region and the expression of *Zdbf2* itself. The authors provide evidence that *Liz* transcription and polycomb function play a minor role in the 3D architecture of the region during differentiation. They end by showing that a CTCF binding site acts to partition the Zbdf2 locus in ESCs. Interestingly, they show that in the absence of this partition, polycomb mediated silencing keeps *Zdbf2* expression at relatively low levels. Removal of both the partition site and polycomb activity results in a dramatic increase of *Zdbf2* expression in ESCs.

The manuscript is well written and robust data generally support the authors' conclusions. The paper will be of interest to the field and will significantly contribute to our understanding of how chromatin organization and architecture change during developmental transitions.

Essential revisions:

1) The state of the cells being examined needs to be better defined. EpiLC differentiation, as originally described by the Saitou lab, typically takes 2-3 days and this recapitulates the postimplantation epiblast (E5.5-6). Some authors refer to this state as formative, rather than primed pluripotency, which in vitro might be best represented by EpiSCs. However, in the manuscript the authors use a 7 days EpiLC differentiation protocol. What kind of cells are the authors really obtaining after 7 days? Are they in fact in the process of adapting their ESC to EpiSC (not EpiLC) conditions? If the typical 2-3 day EpiLC differentiation protocol is considered, then some of the changes in *Zdbf2* expression are not actually observed (E4 deletion or CTCF_PS deletion). Furthermore, it is also unclear whether a 7 day differentiation is always used when referring to EpiLC (e.g. H3K27ac ChIPs). If the authors want to use this 7 day EpiLC differentiation, they should better characterize or describe which cells they are truly working with.

2) The hierarchical activity of *pLiz* and *pZdbf2* needs to be examined in greater detail. There are concerns that clonal differences within the ES cell population could account for some of the described results. This point needs to be addressed in the text and by using another complementary experimental approach such as FISH.

In addition, there is the possibility that E1-3 and E4 enhancers are sequentially used and stage-specific, with E1-3 being important to initially activate the *Zdbf2* proximal promoter (and perhaps E4), while E4 might become dominant in controlling later *Zdbf2* expression. The authors should conditionally delete or inactivate E1-3 (e.g. CRISPRi) during EpiLC differentiation to more conclusively support their model.

3) There are concerns that the *Eed* knockout could have secondary effects. The authors could use an EZH2 inhibitor to more acutely inhibit PRC2 function in their experiments. This would complement their current data and potentially avoid major confounding effects.

4) There are concerns about the authors' claims regarding the CTCF partition. Considering the possibility that loss of CTCF partitioning can lead to a loss of insulation, the authors should evaluate the expression of the *Zdbf2* flanking genes upon deletion of the CTCF PS site in both ESCs and EpiLCs.

In addition, using an alternative experimental approach such as FISH (see point 2 above) to evaluate their various deletion cells would help support their model.

Without CTCF knockout, the authors should temper their language regarding whether CTCF itself is needed for the partition. Alternatively, they could attempt to conditionally knockdown CTCF. This was recently done in ESCs using an AID degree system (Nora et al., 2017) and the authors could perhaps request those cells to save time.

[Editors' note: further revisions were requested prior to acceptance, as described below.]

Thank you for resubmitting your work entitled "Dynamic enhancer partitioning instructs activation of a growth-related gene during exit from naïve pluripotency" for further consideration at *eLife*. Your revised article has been favorably evaluated by Kevin Struhl (Senior Editor), a Reviewing Editor, and two reviewers.

The manuscript has been improved but there are some remaining issues that need to be addressed before acceptance, as outlined below:

The reviewers appreciate that you addressed most of concerns raised during the first round of the review. Reviewer #2 raised a few points that other reviewers agreed to be important points. These points can be simply addressed by textual changes, and we would like you to edit the text to reflect on those points. After that, the reviewing editor should be able to editorially accept the manuscript.

Thank you very much for choosing *eLife* to publish your work. We look forward to receive the revised version soon.

*Reviewer #2:*

I am overall satisfied with how the authors addressed our previous suggestions.

It is true that in some cases they have not been able to solve our concerns due to technical reasons, but it is also true that some of the experiments we proposed were not trivial. In those cases, for example the possibility of E1-3 and E4 enhancers acting in a hierarchical manner, I would suggest that they mention this possibility in their discussion.

They should also incorporate the *Zdbf2* up regulation they observe in the CTCF-AID ESC even if it does not fit their model. I think they can comment on the potential reasons explaining the differences between deleting a single CTCF site and a global loss of CTCF in the Discussion section.

The Author response images 1 A and B should be included as supplementary material, since it helps [to define] what their cells truly are.

*Reviewer #3:*

I agree that the authors more or less addressed the previous suggestions. I also agree that bringing Author response image 1 into the manuscript explains better about the cell status.

---

## [Author Response]

Essential revisions:

*1) The state of the cells being examined needs to be better defined. EpiLC differentiation, as originally described by the Saitou lab, typically takes 2-3 days and this recapitulates the postimplantation epiblast (E5.5-6). Some authors refer to this state as formative, rather than primed pluripotency, which* in vitro *might be best represented by EpiSCs. However, in the manuscript the authors use a 7 days EpiLC differentiation protocol. What kind of cells are the authors really obtaining after 7 days? Are they in fact in the process of adapting their ESC to EpiSC (not EpiLC) conditions? If the typical 2-3 day EpiLC differentiation protocol is considered, then some of the changes in Zdbf2 expression are not actually observed (E4 deletion or CTCF_PS deletion). Furthermore, it is also unclear whether a 7 day differentiation is always used when referring to EpiLC (e.g. H3K27ac ChIPs). If the authors want to use this 7 day EpiLC differentiation, they should better characterize or describe which cells they are truly working with.*

We thank the reviewers for raising this issue, as it is an important point that we did not adequately address in the original manuscript. For the purpose of this study, we wanted to emphasize the promoter switch that occurs at the *Zdbf2* gene. However, in our EpiLC differentiation protocol, the promoter switch does not occur until roughly day 4, and *Liz* is not fully shut down (at the population level) until day 7, which we published in Greenberg et al., 2017. Hence, it was important we used these timepoints in our study. We now justify our strategy in the text, subsection “Two Classes of Putative Enhancers Lie in the *Zdbf2* Locus”, and have added a new panel to that depicts expression dynamics (Figure 1—figure supplement 1B). Furthermore, we clarify throughout the manuscript the day of differentiation for each experiment. We also provide below some RNA-seq analyses that we are not including in the manuscript, but may help answer the reviewer’s question of what cells these are. We performed hierarchical clustering of our ESCs and EpiLCs at various timepoints (in bold italics) with recently published ESC, D2 EpiLC and (in vitro) EpiSC RNA-seq data (Bao et al., 2018) (Author response image 1). Indeed, our D4 and D7 EpiLCs cluster more closely with EpiSCs. Moreover, the primed pluripotency markers Foxa2 and T (Brachyury) are expressed at D7 (Author response image 1). This is also consistent with our in vivo data demonstrating that *Zdbf2* does not activate until E6.5 (Duffié et al., 2014). But without more thorough characterization, we feel it is more prudent continue referring to the cells as D7 EpiLCs.

2) The hierarchical activity of pLiz and pZdbf2 needs to be examined in greater detail. There are concerns that clonal differences within the ES cell population could account for some of the described results. This point needs to be addressed in the text and by using another complementary experimental approach such as FISH.

We agree with the reviewers that clonal variation can lead to misleading interpretations. That said, we are confident in the results presented here, bolstered by some of the experiments performed for this revised version. The enhancer deletion effects are strong on the expression of the genes—in addition to being coherent with the 4C-seq results—that it is highly likely these are regulatory elements. Moreover, experiments in polycomb mutants have been performed in multiple PRC2 genes (Ezh2 in Duffie et al., 2014, *Eed* in Greenberg et al., 2017 and here), as well as in different CRISPR clones for the *Eed* mutation (data not presented). Finally, we used a PRC2 inhibitor as suggested (described in detail below), which was wholly in line with our genetic mutant cell line. The subtlest result was the deletion of the CTCF-binding site. For this revision, we also used the CTCF-degron line, which also points to CTCF-based regulation of *Liz/Zdbf2* (See below).

With regards to performing DNA FISH, this is an experiment we would be eager to perform in principle. However, the locus is quite compact—the distal enhancers (E1-3) and proximal promoter are only separated by about 60kb of genomic space. A DNA FISH experiment would mean implementing a high-resolution microscopy technique that we are not equipped to perform at this moment. We hope the reviewers understand and accept our rationale. Relatedly, we would also like to note that RNA FISH does not allow us to distinguish the *Zdbf2* transcript from the *Liz* transcript, however in our previous publication (Greenberg et al., 2017), we performed single-cell RT-qPCR, which showed that *Liz* and *Zdbf2* promoter activity is mutually exclusive.

In addition, there is the possibility that E1-3 and E4 enhancers are sequentially used and stage-specific, with E1-3 being important to initially activate the Zdbf2 proximal promoter (and perhaps E4), while E4 might become dominant in controlling later Zdbf2 expression. The authors should conditionally delete or inactivate E1-3 (e.g. CRISPRi) during EpiLC differentiation to more conclusively support their model.

We thought this was an absolutely excellent comment, and we were eager to carry out the experiment. We opted for the CRISPRi technique, and designed the experiment thusly: we utilized a doxycycline (dox) inducible Tet-on promoter driving expression of a dCas9-KRAB chimeric construct. We targeted the construct to the ROSA26 locus to ensure proper induction. In parallel we designed a single plasmid containing sgRNAs targeted to E1, E2, and E3 (i.e., three sgRNAs total, each with their own U6 promoter and terminator). Once the dCas9-KRAB line established, we stably transfected the sgRNA plasmid under blasticidin selection. The idea was that we could induce KRAB-mediated silencing of E1-3 at different points during differentiation (Author response image 2), and therefore determine the sequence of events that leads to *Zdbf2* activation. For example, as the reviewers suggested, perhaps we could resolve whether E1-3 are required for maintaining *Zdbf2* expression after the initial activation event. We were able to induce dCas9-KRAB at both the protein and RNA levels (Author response image 2). Unfortunately, in ESCs we did not observe any reduction of the *Liz* transcript upon dCas9-KRAB induction (Author response image 2). This could be due to technical reasons: we did not have time to vet different sgRNAs in the interest of time. Therefore, we simply designed sgRNAs with low off-target scores as close as we could to the center of each enhancer region. However, it is possible that one or more of the sgRNAs did not properly lead to enhancer silencing, and thus *Liz* transcription was unperturbed. We also performed the experiment on a bulk population of sgRNA stable-transfectants. It is possible that clonal sub-populations exhibit silencing, but are in the minority, and the effect was masked. Once again, we were not able to screen clones due to time constraints. Regardless, given the lack of impact on *Liz* in ESCs, we did not feel comfortable carrying out the differentiation experiment without further optimization. In sum, we are quite disappointed we were not able to produce the result for the revision, but we hope the reviewers are sympathetic to the fact that without several more weeks, if not months, of testing we will likely not be able to carry out this particular experiment. As such, we hope the current results using the deletion mutants will be sufficiently convincing to show the hierarchy of events during differentiation.

**Author response image 2. respfig2:** 

3) There are concerns that the Eed knockout could have secondary effects. The authors could use an EZH2 inhibitor to more acutely inhibit PRC2 function in their experiments. This would complement their current data and potentially avoid major confounding effects.

Indeed, we agree with the reviewers that there is always a concern with secondary effects in constitutive mutants, especially after prolonged time in culture. Therefore, as suggested, we incubated both our WT and ∆CTCF_PS ESCs in presence of a potent EZH2 inhibitor (UNC 1999, https://www.tocris.com/products/unc-1999_4904) and a negative control compound (UNC 2400, https://www.tocris.com/products/unc-2400_4905). After 4 days, H3K27me3 was undetectable by western blot, as shown in Figure 4—figure supplement 3D, whereas there was a strong signal in the negative control. RT-qPCR from these cells confirmed our results from the *Eed* mutant cell line, and we have added a panel to Figure 4—figure supplement 3E. This strongly confirms our previous result, and bolsters our interpretation that both polycomb and 3D organization instruct *Zdbf2* activation.

4) There are concerns about the authors' claims regarding the CTCF partition. Considering the possibility that loss of CTCF partitioning can lead to a loss of insulation, the authors should evaluate the expression of the Zdbf2 flanking genes upon deletion of the CTCF PS site in both ESCs and EpiLCs.

This is a good point, and we thank the reviewer’s for raising it. We now have included RT-qPCR graphs for *Eef1b2* and *Ndufs1*, which are immediately upstream of *Liz; Gpr1*, which overlaps *Liz*; and *Adam23*, which is immediately downstream of *Liz/Zdbf2* (Figure 4—figure supplement 2B). There is no significant change in expression for any of these genes during EpiLC differentiation when the CTCF_PS is deleted. Our interpretation of this result is that *Eef1b2* and Ndufs1 remain insulated from the *Zdbf2* locus because of CTCF bound in between *Eed1b2* and *Gpr1* (shown in Figure 4), which does not deplete during differentiation. Our 4C and available HiC data suggest that *Adam23* also exists within its own interacting domain separated from *Zdbf2* possibly because of CTCF-mediated insulation between *Zdbf2* and *Adam23* (Figure 1—figure supplement 2). This result indicates that the CTCF at the partition site plays no role in this insulation. Finally, we have long had interest in *Gpr1*’s possible role in this locus, but we never have observed any major impact on *Gpr1* expression in any of our cis or trans mutations in the ESC-EpiLC system. This experiment is consistent with prior results.

In addition, using an alternative experimental approach such as FISH (see point 2 above) to evaluate their various deletion cells would help support their model.

While we agree, this experiment was too technically challenging to undertake for the purpose of this revision (see above).

Without CTCF knockout, the authors should temper their language regarding whether CTCF itself is needed for the partition. Alternatively, they could attempt to conditionally knockdown CTCF. This was recently done in ESCs using an AID degree system (Nora et al., 2017) and the authors could perhaps request those cells to save time.

We have taken the reviewers’ advice, and moderated our language on the role of CTCF, per se, in the manuscript. Additionally, as suggested, we utilized the CTCF Degron line described in Nora et al. We converted the cells to 2i+vitC (12 days), and then incubated with or without auxin for two days as described in Nora et al. The results are indeed intriguing, and are now incorporated in Figure 4—figure supplement 2D,E. In short, even in the untreated Tir, CTCF-AID line, *Liz* expression is decreased compared to the wild-type control. Our interpretation is that CTCF levels are already lower in the untreated cell line due CTCF protein instability (as previously reported, and confirmed here by western blot). Adding auxin on results in a marginal, statistically insignificant further decrease in expression. Perhaps as even in the untreated cells, lower CTCF levels create a landscape where *Liz* expression is decreased due to decreased enhancer-promoter contacts, and this has reached a minimal threshold. Incidentally, we also observed de-repression of the proximal, *Zdbf2* promoter in both untreated and especially auxin-treated cells. As opposed to our ∆CTCF_PS deletion line where we did not observe a significant effect on *Zdbf2*, globally depleting CTCF apparently leads to upregulation. Whether this is somehow due to direct recruitment of polycomb via CTCF or indirect regulation is a worthwhile avenue of investigation for a future study. As the latter result was somewhat difficult to interpret without further experimental interrogation, we opted to withhold this information in this version of the manuscript.

Finally, we also performed EpiLC differentiation in the absence of CTCF, but we observed so much cell death compared to control cells, we chose not to include those results as we found the data difficult to interpret due to potentially confounding effects.

[Editors' note: further revisions were requested prior to acceptance, as described below.]

The reviewers appreciate that you addressed most of concerns raised during the first round of the review. Reviewer #2 raised a few points that other reviewers agreed to be important points. These points can be simply addressed by textual changes, and we would like you to edit the text to reflect on those points. After that, the reviewing editor should be able to editorially accept the manuscript.Reviewer #2:I am overall satisfied with how the authors addressed our previous suggestions.It is true that in some cases they have not been able to solve our concerns due to technical reasons, but it is also true that some of the experiments we proposed were not trivial. In those cases, for example the possibility of E1-3 and E4 enhancers acting in a hierarchical manner, I would suggest that they mention this possibility in their discussion.

We now discuss a possible hierarchical model of enhancer regulation in the first paragraph of the Discussion.

They should also incorporate the Zdbf2 up regulation they observe in the CTCF-AID ESC even if it does not fit their model. I think they can comment on the potential reasons explaining the differences between deleting a single CTCF site and a global loss of CTCF in the Discussion section.

This figure has been added (Figure 4—figure supplement 2E), and mentioned in the Results subsection “CTCF Partitioning Fine-tunes *pLiz* Programming of *pZdbf2*” and then in the Discussion paragraph two.

The Author response image 1 should be included as supplementary material, since it helps [to define] what their cells truly are.

We now discuss the cell state in subsection “Two Classes of Putative Enhancers Lie in the *Zdbf2* Locus”, and incorporated figure panels (Figure 1—figure supplement 1C,D). We have added the D2 and D4 RNA-seq data to the GEO accession.

Reviewer #3:I agree that the authors more or less addressed the previous suggestions. I also agree that bringing Author response image 1 into the manuscript explains better about the cell status.

See above.